# A Method for Identifying Geospatial Data Sharing Websites by Combining Multi-Source Semantic Information and Machine Learning

**Quanying Cheng** [1,2] , **Yunqiang Zhu** [1,3,*] , **Hongyun Zeng** [4] , **Jia Song** [1,3] , **Shu Wang** [1] , **Jinqu Zhang** [5] , **Lang Qian** [5] **and Yanmin Qi** [1]

1   State Key Laboratory of Resources and Environmental Information System, Institute of Geographic Sciences and Natural Resources Research, Chinese Academy of Sciences, Beijing 100101, China; chengqy.18b@igsnrr.ac.cn (Q.C.); songj@lreis.ac.cn (J.S.); wangshu@igsnrr.ac.cn (S.W.); sam_qi@foxmail.com (Y.Q.)
2   University of Chinese Academy of Sciences, Beijing 100049, China
3   Jiangsu Center for Collaborative Innovation in Geographical Information Resource Development and Application, Nanjing 210023, China
4   School of Earth Sciences, Yunnan University, Kunming 650500, China; hy_zeng@ynu.edu.cn
5   School of Computer Science, South China Normal University, Guangzhou 510000, China; zjq@scnu.edu.cn (J.Z.); 2018022623@m.scnu.edu.cn (L.Q.)
*   Correspondence: zhuyq@lreis.ac.cn

**Abstract:** Geospatial data sharing is an inevitable requirement for scientific and technological innovation and economic and social development decisions in the era of big data. With the development of modern information technology, especially Web 2.0, a large number of geospatial data sharing websites (GDSW) have been developed on the Internet. GDSW is a point of access to geospatial data, which is able to provide a geospatial data inventory. How to precisely identify these data websites is the foundation and prerequisite of sharing and utilizing web geospatial data and is also the main challenge of data sharing at this stage. GDSW identification can be regarded as a binary website classification problem, which can be solved by the current popular machine learning method. However, the websites obtained from the Internet contain a large number of blogs, companies, institutions, etc. If GDSW is directly used as the sample data of machine learning, it will greatly affect the classification precision. For this reason, this paper proposes a method to precisely identify GDSW by combining multi-source semantic information and machine learning. Firstly, based on the keyword set, we used the Baidu search engine to find the websites that may be related to geospatial data in the open web environment. Then, we used the multi-source semantic information of geospatial data content, morphology, sources, and shared websites to filter out a large number of websites that contained geospatial keywords but were not related to geospatial data in the search results through the calculation of comprehensive similarity. Finally, the filtered geospatial data websites were used as the sample data of machine learning, and the GDSWs were identified and evaluated. In this paper, training sets are extracted from the original search data and the data filtered by multi-source semantics, the two datasets are trained by machine learning classification algorithms (KNN, LR, RF, and SVM), and the same test datasets are predicted. The results show that: (1) compared with the four classification algorithms, the classification precision of RF and SVM on the original data is higher than that of the other two algorithms. (2) Taking the data filtered by multi-source semantic information as the sample data for machine learning, the precision of all classification algorithms has been greatly improved. The SVM algorithm has the highest precision among the four classification algorithms. (3) In order to verify the robustness of this method, different initial sample data mentioned above are selected for classification using the same method. The results show that, among the four classification algorithms, the classification precision of SVM is still the highest, which shows that the proposed method is robust and scalable. Therefore, taking the data filtered by multi-source semantic information as the sample data to train through machine learning can effectively improve the classification precision of GDSW, and comparing the four classification algorithms, SVM has the

best classification effect. In addition, this method has good robustness, which is of great significance to promote and facilitate the sharing and utilization of open geospatial data.

**Keywords:** geospatial data sharing website; multi-source semantic information; website classification; natural language processing technology; machine learning

## 1. Introduction

The development of technology in earth observation, ground monitoring, deep detection, and social network and mobile voluntary data collection has boosted the approaches to obtain geospatial data. Rich geospatial data can better support research on natural resources, ecological environment change and its sustainable development, and national strategic decision-making [1]. To promote the acquisition and wide use of geospatial data, a large number of geospatial data are available to the public through portals or websites online by government departments, research institutions, or enterprises [2,3]. Geospatial data portals or websites (hereafter referred to as geospatial data sharing website, GDSW) refer to the human-to-machine interface performing as a single point-of-access to geospatial data. It is the bridge between geospatial data providers and users, and typically employed as a web-based graphical user interface (GUI) equipped with functionalities for accessing geospatial data [4,5].

At present, the main approach to sharing geospatial data on the web is to access GDSW directly [6–8], such as NASA's Earth Observing System Data and Information System (EOSDIS), Global Change Master Directory (GCMD, http://gcmd.nasa.gov, last accessed: 15 September 2021), and Global Land Analysis & Discovery (GLAD, https://glad.umd.edu/dataset, last accessed: 15 September 2021) [9]. However, users are currently limited to using datasets provided by data websites they are familiar with instead of the ones that are more suitable for their requirements [9–11]. The main reason is that, so far, no websites have been found to provide a comprehensive index of all geospatial data [11]. Although there are integrated data websites, such as the Global Change Master Directory (GCMD), the data provided by these are not enough, because the data on this website mainly come from various departments of National Aeronautics and Space Administration (NASA), and there are other websites that contain a lot of geospatial data. Additionally, the current GDSWs hosted by different organizations have various forms of data release—for example, they are published based on different standards—which leads to the isolated existence of GDSW on the web, thus bringing great challenges for users to discover geospatial data for research and decision support applications [12,13]. Therefore, the precise discovery of GDSWs is the foundation and prerequisite for precisely mining and utilizing open geospatial data, which can greatly save scientists' time in finding geospatial data, bring great convenience to researchers and related personnel, and effectively improve the utilization of geospatial data.

There are two types of GDSW: one is based on the open geospatial consortium (Inc., OGC), that is, an open-standard data website [14], and the other one is non-open-standard [15], which does not depend on OGC standard when publishing geospatial data. According to the published GDSW based on the OGC standard, whether the website is a geospatial data website can be evaluated by sending a specific request (see [16] for details). Although the open-standard website can identify geospatial data websites precisely and quickly using a simple method, it cannot find data websites that are not published based on OGC standards. There are a large number of non-standard geospatial data websites in the network [8], which illustrates that the method is incomplete in identifying GDSW. For non-open-standard websites, users search through keywords, and search engines use precise or fuzzy matching to search the whole network and return a series of results [17], and then identify whether there are GDSWs in the results one by one. However, the results returned by traditional search engines often contain a large amount of irrelevant information, such

as companies, blogs, institutions, etc., which is time- and energy-consuming for users to identify. Some scholars use semantics to enhance the recognition ability of the above two types of GDSW [18,19]. Using semantics can improve the recognition ability of GDSW, since this method relies on the semantic library, yet the existing semantic library is only the semantic information constructed in a certain field, such as expanding discovery by using the semantic web for earth and environmental terminology (SWEET), expanding discovery using ocean ontology, and so on [20–22]. Therefore, there are still great limitations in using existing methods to identify GDSW.

The recognition of GDSW can be regarded as the classification of 'text documents' (here, they are websites) [23–25]. In this case, this task is also called website categorization or website classification (see [26]). Currently, machine learning methods are frequently used to solve classification problems [27–30] and can be divided into supervised classification and unsupervised classification. Supervised classification has been widely used because it can make full use of prior knowledge to learn feature information, which includes using a group of 'text documents' (each document has a class label) to learn about the classifier, and then using the classifier to automatically assign the class label to the new unmarked 'text document'. However, the website is not standardized and provides information, such as text, to users in various forms, which means the data obtained by automatic scraping contain noise. To reduce such dominant noise content, topic extraction technology that can extract the main information from the data obtained by automatic scraping is required. In addition, a filtering method is required to first filter the existing irrelevant websites from the acquired data.

In this paper, a method of website classification is proposed to cope with the above-mentioned difficulties. The method is based on automatic text scraping and topic extraction technology to create data records representing websites, uses semantic technology to filter out a large number of irrelevant data records, performs classification based on classification algorithms (KNN, LR, RF, and SVM), and finally, analyzes and evaluates the obtained website classification results. The rest of this paper is organized as follows: Section 2 introduces the overall framework of the proposed method. It mainly includes the acquisition and preprocessing of GDSW, the data filtering based on multi-source semantic information, and the classification algorithms and evaluation methods used in the classification phase. The classification results are displayed and evaluated in Section 3. Section 4 discusses the method proposed in this paper and summarizes the conclusion.

## 2. Methodology

### 2.1. Overall Framework

The overall GDSW identification method consists of four steps, as shown in Figure 1. (1) Original website acquisition and pre-processing. We used a series of keywords related to geospatial data to obtain the original website list L through the Baidu search engine. A large number of websites in the website list exist in the same domain name, so we performed the domain name extraction first. Then, we downloaded the website documents and performed feature extraction to obtain the main information of the website. (2) Geospatial data website filtering based on multi-source semantic information. We calculated the similarity value between multi-source semantic information and websites to remove unrelated websites and obtain the potential geospatial data website list W. Multi-source semantic information contains geospatial data semantics and website feature semantics. The geospatial data semantics include geospatial data content ontology, source ontology, and morphology ontology (see Section 2.3 for details). (3) Precise identification of GDSW based on the machine learning method. With website lists L and W as sample data, respectively, we used machine learning [31] for classification and identification. The data were manually labeled before classification. (4) Evaluation and analysis of the geospatial data website identification precision. We designed two comparative experiments to evaluate the precision, recall, F1-score, precision–recall, and receiver operating characteristic curve (ROC curve) of classification results, using original data and multi-source semantic information filtering.

The first experiment divided L into geospatial data sharing websites (GDSW) and non-geospatial data sharing websites (non-GDSW). In GDSW and non-GDSW, we selected the data as training sets according to a certain proportion, respectively, and used the rest as test sets. W was also divided into training set and test set according to the same method and used for the second experiment. To fully verify the robustness and generalizability of applying multi-source semantic information to solve the imbalance of sample data, we conducted the third set of experiments and evaluated the results using different sample data from those mentioned above.

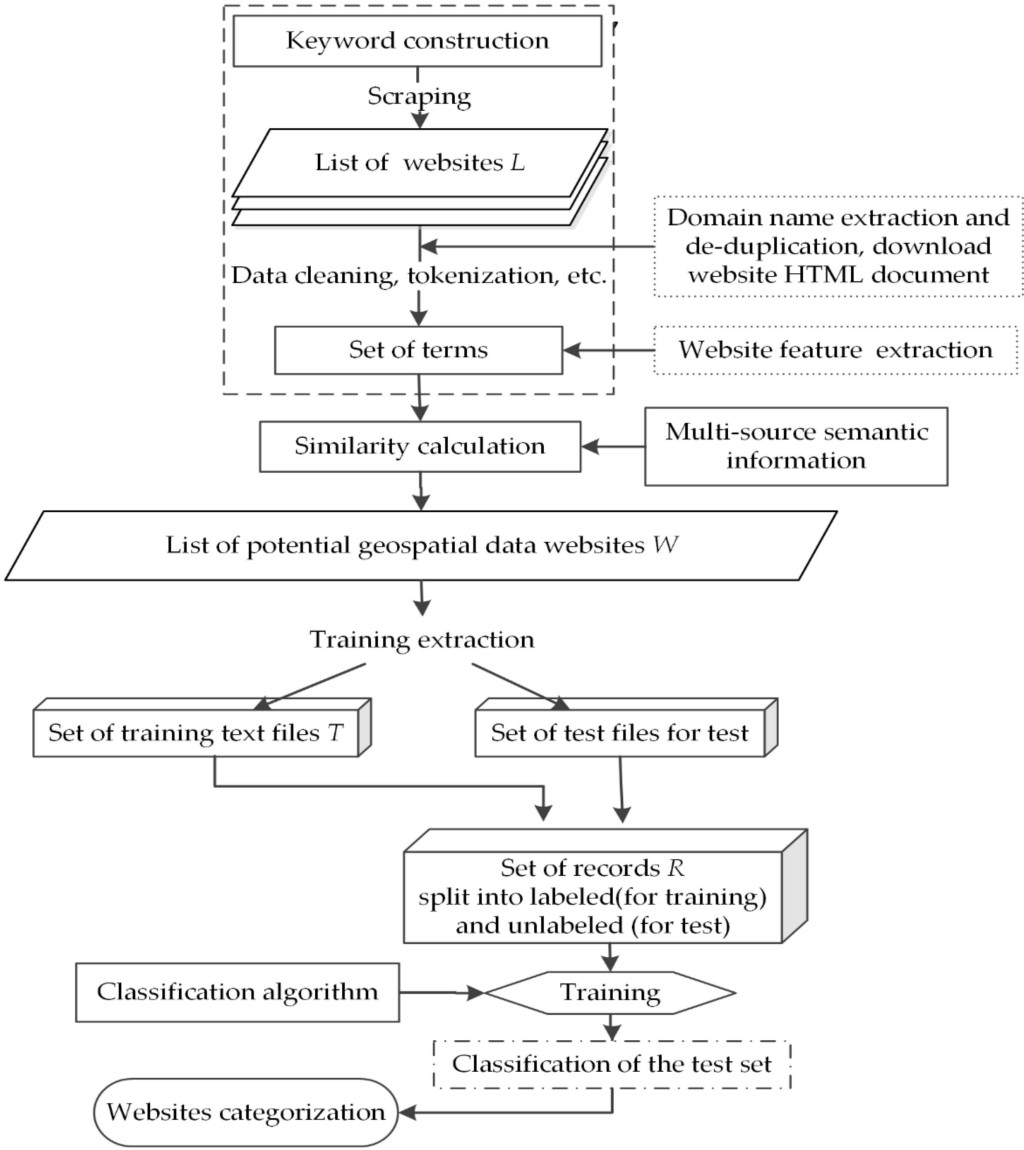

**Figure 1.** GDSW identification method.

### 2.2. Data Source and Preprocessing

There are five steps for data acquisition and pre-processing, as shown in Figure 2. (1) Keyword construction; (2) keyword-based website acquisition; (3) domain name extraction and de-duplication; (4) text data pre-processing—the HTML document of each website was downloaded and processed, and special symbols, word segmentations, stop words, etc. were removed; (5) website feature information extraction. Because the texts contained in websites are very complicated, a geospatial data website, for example, may contain a lot of news information, so extracting the features of each website is key. Therefore, we use

text keyword extraction technology to extract website features, and finally obtain the key information of each website. See Sections 2.2.1 and 2.2.2 for details.

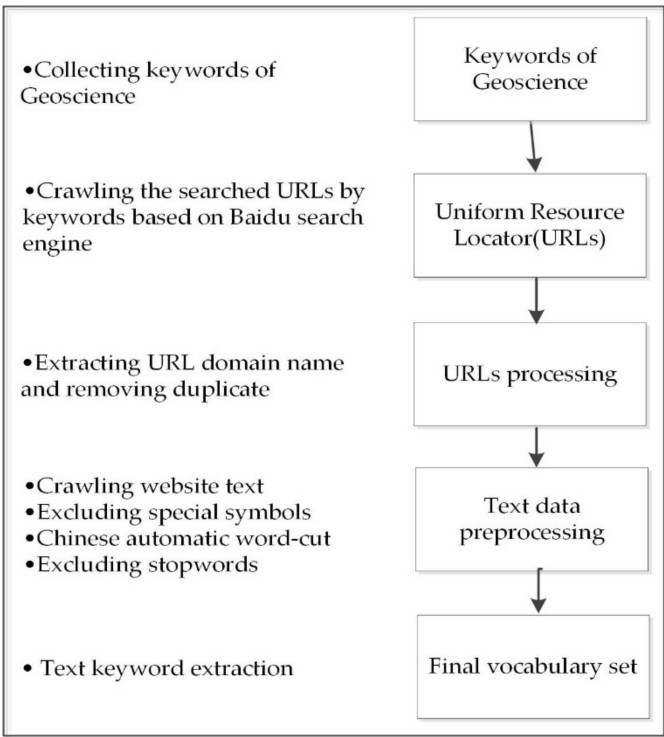

**Figure 2.** Keyword-based GDSW acquisition and pre-processing process.

2.2.1. Website List Extraction

The list of GDSW consists of two parts: the URLs directly searched by the Baidu search engine based on keywords and the associated URLs embedded in the webpage text of search results, as the URLs in the webpage text are highly likely to be GDSW.

The first part of URL acquisition requires the construction of keywords. In total, 70 keywords were selected from the second classification of the keywords table of Earth system science data in the literature [32], such as ground remote sensing, land use, precipitation, etc. Next, each keyword was set as the input information of the search engine, and the search engine could crawl many URLs with the assistance of the automatic crawler (note: the search engine is a fuzzy matching query based on the input keywords). The process is shown in Figure 3.

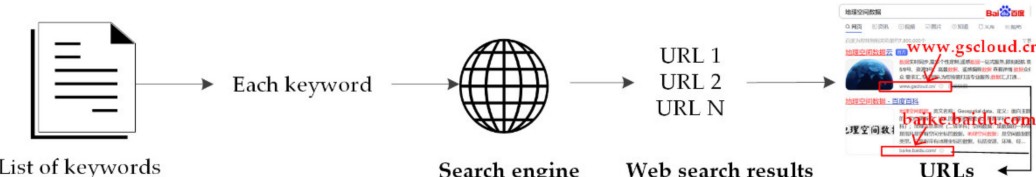

**Figure 3.** The basic flow of website list generation.

The second part is to identify the URLs embedded in the webpage text. Many URLs appear in the anchor element (<a>) of the HTML document, and the crawler extracts URLs from the "href" attribute of the anchor element. In addition, URLs may also appear in the plain text of web pages, and these URLs cannot be extracted by the href attribute. The regular expression is designed to discover URLs in the plain text of webpages [11].

As the obtained URLs may come from different pages of the same website, it is necessary to extract the domain name from the URL list. Therefore, a domain name extraction algorithm was designed and implemented.

### 2.2.2. Pre-Processing and Feature Extraction

The downloaded HTML document extracts the theme and content parts. The theme part is the header information in the HTML document, which can characterize the type of website, and the content part comes from the main part of the web document. The theme information was extracted from the <head></head> tag. Besides the page title described by the <title> tag, the content attributes in <meta name = "description" content = ""> and <meta name = "keywords" content = ""> also describe important information related to the theme [33,34]. The content information is extracted from the <body> tag.

It is necessary to preprocess the extracted website information. First, we designed a series of regular expressions to remove the meaningless information, such as tags and symbols. After that, we applied the word segmentation tool, since the research content in Chinese does not contain blank characters. In this study, we used an open-source word segmentation tool (https://github.com/fxsjy/jieba, last accessed: 15 September 2021) named jieba library. To get website feature information, we selected the TF–IDF (term frequency–inverse document frequency) algorithm from the jieba library to implement website keyword extraction, and the key parameter was topK; that is, the number of keywords that were extracted as features. In the study, the value of this parameter was set to 50, because the features obtained from the larger range contained too much noisy information. The principle of jieba library and the TF–IDF algorithm is as follows.

Jieba library is a famous Chinese lexical analysis system that adopts the model based on HMM (Hidden Markov Model) and contains several algorithms, among which are the Viterbi algorithm [35] for unknown words, the textrank algorithm [36], and the TF–IDF [37] algorithm for keyword extraction.

Term frequency–inverse document frequency (TF–IDF) is a weighting technique commonly used in the field of information retrieval to measure the importance of a word in a document or corpus. TF–IDF converts a text corpus into a numerical vector, where each number represents the importance of each word for a given document [38]. The importance of a word increases proportionally with its number of occurrences in the document but decreases inversely with its frequency in the corpus. The numerical score is calculated by multiplying the term frequency (that is, the number of times a word appears in a document) by the reciprocal of the document frequency (Equation (1)).

$$w_{i,j} = tf_{ij} \times \log(\frac{N}{df_i}) \tag{1}$$

where $tf_{ij}$ is the frequency of $i$ appearing in $j$, $df_i$ is the number of documents containing $i$, and $N$ is the total number of documents.

### 2.3. Data Filtering Based on Multi-Source Semantic Information

The pre-processed websites contain a large number of blogs, companies, institutions, etc., which are irrelevant with geospatial data. The main reason is that the search engine performs fuzzy matching according to the input keywords and returns all the contents that contain full or partial keywords, which leads to a large number of non-geospatial data websites in the search results. Therefore, when we use machine learning methods to learn the features of geospatial data websites and non-geospatial data websites, the difference between the features of the two cannot be fully learned, which will affect the classification precision. Therefore, before using the machine learning method to precisely identify GDSW, we first filtered the pre-processed website list with semantic information.

The similarity calculation based on multi-source semantic information mainly has three steps, as shown in Figure 4. (1) Establishment of multi-source semantic information. Geospatial data semantics and website feature semantics were constructed. The geospatial

data semantics included morphological semantics, source semantics, and content semantics. (2) Generation of word vector. Before the similarity calculation, both website text and multi-source semantic information needed to be represented in vectors. In this study, a word2vec neural network model was trained to obtain a vector representation of words. (3) Similarity calculation. The similarity calculation included two parts, as shown in Part I and Part II in Figure 4. First, the semantic similarity between website theme and website feature semantics was calculated. If the similarity value was greater than a certain threshold, it is considered a potential geospatial data website. If the similarity value could not be judged, the comprehensive similarity between website and geospatial data content ontology (GDCO), geospatial data morphology ontology (GDMO), and geospatial data source ontology (GDSO) were calculated, that is, Part II in Figure 4, and the comprehensive similarity was used to judge again.

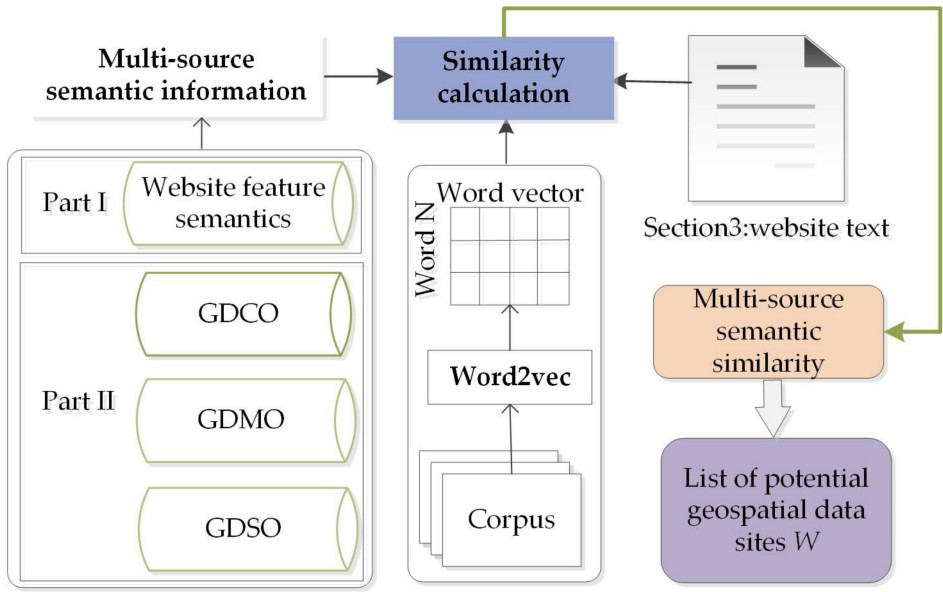

**Figure 4.** Flow chart of website filtering based on multi-source semantic information.

Multi-source semantic information contains geospatial data content and geospatial data website characteristics. If a website is a geospatial data website, it must have metadata information of geospatial data and certain descriptive information, such as a data platform and sharing center. Therefore, the semantic characteristics of these two features can effectively filter non-geospatial data websites.

As a formal expression of semantics, ontology plays an important role in semantic computing. The morphological ontology of geospatial data is a formal specification description of the conceptual system of morphological characteristics of geospatial data sharing, which contains descriptive information such as data type, data structure, data format, and storage medium. The source ontology of geospatial data is a formal specification of the source concept of geospatial data sharing, containing descriptive information such as data collection and data processing. Both morphological ontology and source ontology use existing achievements [39,40].

A geospatial data content ontology (assumed as O) was constructed and divided into two categories at the conceptual level, including basic geographic elements and thematic elements. Ontology O was established using protégé [21], an open-source software, and mainly used for knowledge-based ontology construction and ontology application in the semantic web. The content ontology of geospatial data includes 624 concepts and 470 individuals. Note that all the information contained in ontology in this study will be used as keywords to calculate similarity with websites.

The semantics of website features mainly measure the degree of semantic similarity of the website theme text (i.e., theme information in Section 2.2.2). The semantic library contains two aspects: (1) keywords from the primary classification of the earth system science data keyword list in the literature [32], which mainly reflects the category of websites; and (2) some self-defined feature vocabularies of GDSW. For example, the <title></title> tag of the website usually contains words such as data center and data sharing, which are unified as theme words. In addition to the above-mentioned geospatial data semantics and website feature semantics, the URL itself can also be filtered by certain rules, such as Baidu Encyclopedia, blog, etc. Therefore, links unrelated to geospatial data website URLs were defined to exclude irrelevant URLs.

### 2.3.1. Word Vector Generation

Before the similarity calculation, the data should be transformed into vector representation. In this study, we used the word2vec model, a neural network model that represents words in the corpus as vectors with contextual understanding [41]. In vector space, the closer the distance between two vectors is, the more similar the two words are. The results of word2vec depend on two user-defined parameters: the dimension of vector space (i.e., size) and the maximum distance between a word and the surrounding words in a sentence (i.e., window). Word2vec can be configured in two ways: skip-gram and the continuous bag of words (CBOW). The main difference is that skip-gram aims to predict the surrounding words by entering reference words, while CBOW uses the surrounding words to predict the current words. In this study, we set the dimension and window of vector space to 100 and 5, respectively, and selected CBOW as the configuration mode. Finally, we obtained the n-dimensional vector space and calculated the similarity between words using the distance function, which, in turn, calculates the similarity between multi-source semantic information and websites.

### 2.3.2. Similarity Calculation

1.  Calculation method:

The semantic similarity between words was quantified by the distance function. It was assumed that $\vec{k} = (k_1, \ldots, k_n)$ and $\vec{k'} = (k'_1, \ldots, k'_n)$ are two-word vectors of website and multi-source semantic information, respectively. $\vec{k}$ and $\vec{k'}$ use the cosine function of N-dimensional vector space to calculate their similarity, and the distance function should satisfy the following four axioms [42]:

*   $f(x, y) \geq 0 \, (non-negativity)$
*   $f(x, y) = 0 \, (identity)$
*   $f(x, y) = f(y, x) \, (symmetry)$
*   $f(x, y) \leq f(x, y) + f(y, z) \, (subadditivity)$

Here, the parameters $x, y, z$ are members of n-dimensional vector space.
The cosine distance is described as follows:

$$d_{cos}(\vec{k}, \vec{k'}) = \frac{\sum_{i=1}^{n} |k_i \times k'_i|}{\|\vec{k}\| \times \|\vec{k'}\|} \tag{2}$$

where $\|\vec{k}\|$ is $\sqrt{\sum_{i=1}^{n} (k_i)^2}$, and $\|\vec{k'}\|$ is $\sqrt{\sum_{i=1}^{n} (k'_i)^2}$.

The similarity between words is calculated, and the similarity of word sets is calculated as follows: in word sets A and B, A has m words, namely $A_1, \ldots, A_m$, where $A_1$ is the first word of A, and $A_m$ is the m-th word of A; similarly, B has n words, namely $B_1, \ldots, B_n$, and $B_n$ is the n-th word of B; then, the similarity between $A_m$ and $B_n$ is $Sim(A_m, B_n)$, and the similarity can be calculated for each word in A and each word in B. First, we compared $A_1$ with $B_1$ to $B_n$ to find the similarity value of the words with the largest similarity

value, then compared $A_2$ with $B_1$ to $B_n$ to find the similarity value of the words with the highest similarity between $A_2$ and B. Finally, we summed these maximum similarity values, divided this sum by the average length of two-word sets, and obtained the sentence similarity between A and B. The formula is as follows [42]:

$$Sim(A, B) = \frac{\sum_{i=1}^{m} Max(Sim(A_i, B_1), \ldots, Sim(A_i, B_n))}{L}, \; L = L(A) + \frac{|L(B) - L(A)|}{2} \quad (3)$$

In the formula, the length of A is $L(A)$, and the length of B is $L(B)$.

According to the similarity calculation formula, we calculated the similarity between each website in L and the multi-source semantic information. First, we calculated the website features, that is, the semantic similarity between the theme of each URL and the preset website features. If the similarity was greater than 0.9, it was considered a potential geospatial data website and stored in W. When we could not determine whether it is a potential geospatial data website through the website features, the GDCO, GDMO, and GDSO were applied to calculate the data similarity with the website content, respectively. Finally, we obtained the comprehensive similarity (as in Equation (4)). If the comprehensive similarity was greater than 0.8, it was considered a potential geospatial data website and stored in W. The similarity thresholds of website features and data features were set to 0.9 and 0.8, respectively. Through the statistical analysis, clustering, and manual assessment, we found that, when the similarity thresholds of website features and data features were less than 0.9 and 0.8, respectively, the corresponding websites were almost non-geospatial data websites.

$$\text{TotalSim} = \alpha \times \text{GDCO} + \beta \times \text{GDMO} + \gamma \times \text{GDSO} \quad (4)$$

where $\alpha$, $\beta$, and $\gamma$ are the trade-off parameters between the website and geospatial data content ontology, morphology ontology, and source ontology; the sum of the three cannot exceed 1, and the value is determined by the analytic hierarchy process (AHP) [43], which are 0.67, 0.22, and 0.11, respectively. Before selecting those weights, the influence analysis was carried out according to each similarity value.

2.  Calculation method:

The Algorithm 1 for calculating the similarity of websites and multi-source semantic information is as follows.

We applied the described process to a list of 79,402 URLs, obtained 2760 original websites after domain name extraction and deduplication, and isolated 638 after using multi-source semantic information as the filter. Finally, we used the 2760 and 638 websites as sample data for machine learning to perform the training and testing phases.

### *2.4. Classification of GDSW Based on Machine Learning*

#### 2.4.1. Classification Algorithm

Identifying GDSW can be transformed into a dichotomous classification problem in essence. Currently, many different classification methods have been proposed in the literature based on different data models and data technologies. It is commonly acknowledged that no method can outperform the others in all cases. However, given a specific classification problem, it is empirically possible to determine which methods usually provide the best performance for that class of problem. In this study, we performed different classifiers using a very good machine learning package at present called the scikit-learn library [44]. For the classification of GDSW, we tested several classifiers preliminarily, and obtained preliminary results in the following classifiers:

(1)    Support Vector Machines (SVM);
(2)    K-Nearest Neighbor (KNN);
(3)    Random Forests (RF);
(4)    Logistic Regression (LR).

---

**Algorithm 1** Filtering algorithm.

| | |
|---|---|
| **Definition**: | |
| 1: | $URL_S$: A series of URLs based on keywords crawled by Baidu search engine. |
| 2: | $w$: Text that represents the website. |
| 3: | $w_{head}$: The text in the <head></head> tag in the HTML document of the website, the text has been processed by word segmentation, stop words removal, theme word extraction (TF–IDF algorithm extraction), etc. See Sections 3.1 and 3.2 for details. |
| 4: | $\overrightarrow{w_{head}}$ : The vector representation of $w_{head}$. |
| 5: | $w_{body}$ : The preprocessing method is the same as $w_{head}$, but for text in the < body></body> tag. |
| 6: | $\overrightarrow{w_{body}}$: The vector representation of $w_{body}$. |
| 7: | $un_u$ : Customized domain name libraries uncorrelated to GDSW. |
| 8: | $T$ : Website feature semantics. |
| 9: | $\overrightarrow{theme}$ : The vector representation of website feature semantics. |
| 10: | $C, M, S$: Semantic information composed of GDCO, GDMO, and GDSO, respectively. |
| 11: | $\overrightarrow{c_{onto}}$, $\overrightarrow{m_{onto}}$, $\overrightarrow{s_{onto}}$: The vector representation of $C, M, S$, respectively. |
| 12: | $sim_{head}$: The similarity value between $w_{head}$ and theme, which is initialized to 0. |
| 13: | $sim_{body}$: The comprehensive similarity values calculated by $w_{body}$ and GDCO, GDMO, and GDSO, which is initialized to 0. |
| 14: | $FilterURL_S$ : List of websites storing potential geospatial data websites. |
| **Input**: | $URL_S$ |
| **Output**: | $FilterURL_S$ |
| 15: | $URL_{dm} \longleftarrow URL_S$ /*Extract the domain names of all URLs and de-duplicate them to get the list of websites $URL_{dm}$*/ |
| 16: | **for** $l_i$ **in** $URL_{dm}$**:** |
| 17: | **for** url **in** $un_u$ |
| 18: | **if** url **in** $l_i$: /* According to the characteristics of the URL itself, if it contains the set domain names, such as Baidu Encyclopedia, it will be deleted directly from the database */ |
| 19: | **pass** |
| 20: | **elif** $sim_{head} > 0.9$: /* $sim_{head} \longleftarrow$ T $\left( \overrightarrow{w_{head}}, \overrightarrow{theme} \right)$*/ |
| 21: | $W_{store}$ /* Save URLs with theme similarity value greater than 0.9 into $FilterURL_S$. */ |
| 22: | **elif** $sim_{body} > 0.8$ : /*$sim_{body} \longleftarrow C\left( \overrightarrow{w_{body}}, \overrightarrow{c_{onto}} \right), M\left( \overrightarrow{w_{body}}, \overrightarrow{m_{onto}} \right), S\left( \overrightarrow{w_{body}}, \overrightarrow{s_{onto}} \right)$, comprehensive similarity between $w_{body}$ and C,M,S */ |
| 23: | $W_{store}$ /* Save URLs with comprehensive similarity value greater than 0.8 into $FilterURL_S$. */ |
| 24: | **else:** |
| 25: | **pass** |
| 26: | **return** $FilterURL_S$; |

---

The SVM algorithm divides the n-dimensional spatial representation of data into two regions using a hyperplane, which always maximizes the boundary between these two regions or classes. The boundary is defined by the farthest distance between the instances of two classes and is calculated according to the distance between the nearest instances of two classes, which are called support vectors [44]. Then, the new examples are mapped to the same space, and they are predicted to belong to a category according to which side of the hyperplane they fall on. In addition to performing linear classification, SVMs can use kernel functions to perform nonlinear classification surfaces, such as polynomial surfaces, radial surfaces, or sigmoid surfaces [38,45,46].

KNN is one of the simplest supervised machine learning models. The basic idea is to classify unknown samples according to the closest instance category in the training space by measuring the distance between training instances and unknown instances. So, the most important parameter of this algorithm is the distance function, including Euclidean

distance, Manhattan distance, Markov distance, etc. The KNN algorithm is easy to use and insensitive to outliers. However, KNN is a lazy algorithm, which is easy to build models with, but is costly in the system of classifying test data [47].

RF refers to a classifier that uses multiple trees to train and predict samples. It is an ensemble learning method where the goal is to create a decision tree to predict the value of target variables according to the combination of input variables. Each internal node is associated with a decision related to the value of the input variable, which can best segment the training set. Different algorithms can be used to determine the input variables associated with internal nodes; see also [48]. The global output is obtained by calculating the output mode of each tree, and more details can be found in [49,50]. RF is generally robust, can achieve better performance than a single decision tree, and can be extended to deal with very large datasets (for example, see [51]). The number of trees used in the experiment is set to 300.

LR is a regression model in which the target variables are classified. Therefore, it can be used for classification. It measures the relationship between the target variable and one or more independent variables by estimating the probability with a logical function, that is, a cumulative logical distribution; see also [52]. LR can be regarded as a special case of the generalized linear model, so it is similar to a linear regression. This method is often used in practice because it has less computation and considerable generalization ability.

When using a supervised classification method in machine learning, we need to label the sample data before implementation. If the website provides geospatial data, we labelled it as positive; otherwise, we labelled it as negative. Then, we divided the sample data into the training set and test set according to a certain proportion. We selected the positive class and negative class in the training set according to the proportion of 2/3~4/5 of their respective numbers. Subsequently, we used the training set as the input information of machine learning to perform the training stage, and after the training was completed, we tested the test set. Finally, we calculated the classification result according to the predicted category and the true category.

### 2.4.2. Evaluation Methodology

When the model training was completed, we used the classifier to predict the categories of all records in the test set and compare the predicted categories with the real categories, so that we could calculate the corresponding confusion matrix and later use the elements of each confusion matrix (true positives *TP*, false negatives *FN*, true negatives *TN*, and false positives *FP)* to evaluate the performance. In this paper, we evaluate the precision, recall, and F1-score. The specific calculation method is as follows:

- *Precision*, also known as positive prediction value, is defined as the percentage of true positive records among all positive predictions:

$$Precision = \frac{100\ TP}{TP + FP} \tag{5}$$

- *Recall*, also known as true positive rate, is defined as the percentage of true positive predictions in all true positive records:

$$Recall = \frac{100\ TP}{TP + FN} \tag{6}$$

- *F1-score* is the harmonic average of precision and recall:

$$F1\text{-}score = \frac{200TP}{2TP + FP + FN} \tag{7}$$

The precision and recall are mutually restrained and contradictory, which cannot be increased at the same time. When the threshold value increases, the precision increases, and the recall decreases. When the threshold value decreases, the precision decreases, and the

recall increases. The precision–recall curve is a curve calculated by changing the threshold with recall as x axis and precision as y axis, which can better reflect the real performance of classification when the ratio of positive and negative samples is quite different.

The ROC curve can reflect the sensitivity and precision of the model when choosing different thresholds. When the distribution of positive and negative samples changes, the shape of the curve can remain unchanged. The curve contains two important values: one is the true positive rate (TPR), and the calculation formula is TPR = $TP/(TP + FN)$, which describes the proportion of positive instances identified by the classifier to all positive instances; the other is the false positive rate (FPR), calculated as FPR = $FP/(FP + TN)$, which is the proportion of negative instances that the classifier incorrectly identifies as positive to all negative instances. In the ROC graph, each point is drawn with the corresponding FPR value as the horizontal coordinate and TPR value as the vertical coordinate.

The ROC curve needs a quantitative analysis of the model. Here, the area under the ROC curve (AUROC) needs to be introduced. AUROC refers to the area below the ROC curve. To calculate AUROC, it only needs to integrate along the horizontal axis of ROC. In real scenes, the ROC curve is generally above the y = x line, so the value of AUROC is generally between 0.5 and 1. The larger the AUROC value is, the better the performance of the model will be.

## 3. Results and Evaluation

First, we manually marked 2760 original websites. If the corresponding websites provided geospatial data, we marked them as positive; otherwise, we marked them as negative. To perform the classification task, we selected 50% of the records in the datasets as training sets, and the rest as test sets. After that, we trained the classifiers using different classification algorithms and tested the test sets using the classifiers. We randomly performed the extraction of training sets three times, respectively, and averaged all performance results on the three trials (the first set of experiments). Similarly, we manually marked 628 websites filtered by multi-source semantic information, and obtained and trained the training set by the same method, but the prediction set was the same as that mentioned above to verify the effectiveness of multi-source semantic information for classification (the second set of experiments). See Section 3.1 for details.

To verify the robustness of the proposed method, we obtained the second set of data using different keyword groups based on the Baidu search engine and used the same method for training and evaluation, as described in Section 3.2.

### 3.1. Evaluation of Classification Results Based on Machine Learning

We performed the training phase using KNN, LR, RF, and SVM for the above training set, and set the main parameters of each classifier as shown in Table 1. After the training phase, we used the learned classifier to predict the categories of all records in the test set and compared them with the real categories to calculate the confusion matrix. Table 2 shows the classification precision, recall rate, and F1-score calculated according to the confusion matrix. It can be seen that, when the pre-processed sample data are used directly for classification, the highest precision can reach up to 80% using RF and SVM. For the same classification precision of utilizing RF and SVM, the recall and F1-score of SVM are higher.

**Table 1.** Main parameter setting of KNN, LR, RF and SVM.

| KNN | LR | RF | SVM |
|---|---|---|---|
| n_neighbors = 35 | max_iter = 1100 | n_estimators = 100 | Kernel = 'linear' |

**Table 2.** *Precision*, *recall*, and *F1-score* of KNN, LR, RF, and SVM on pre-processed sample data.

| Classification Algorithm | Precision | Recall | F1-Score |
|:---:|:---:|:---:|:---:|
| KNN | 57.9% | 34.9% | 43.6% |
| LR | 74.4% | 50.8% | 60.4% |
| RF | 80% | 31.7% | 45.5% |
| SVM | 80% | 44.4% | 57.1% |

After that, we trained the second group of experimental data using the four classification methods, and predicted the test set in the first group of experimental data. The results show that (as shown in Table 3) the classification precision was greatly improved in all four classification algorithms compared with using the sample data that were only pre-processed. The classification precision (95%) and the corresponding F1-score (58.5%) of SVM are the highest, accordingly. Combined with the precision–recall (PR) curve (as shown in Figure 5), it can be seen that, overall, the classification precision of SVM is also better with consistent recall, and the classification precision of LR is relatively higher only when recall is greater than 0.75. Meanwhile, for the ROC curve (as shown in Figure 6), the classification performance of all four classification algorithms is good, and their AUROC values are all floating around 0.9 with little fluctuation. The AUROC value of LR (0.93) is the highest, followed by SVM (0.92). When the classification performance is good in all cases, we usually select the classifier with higher classification precision because our goal is to precisely identify GDSW.

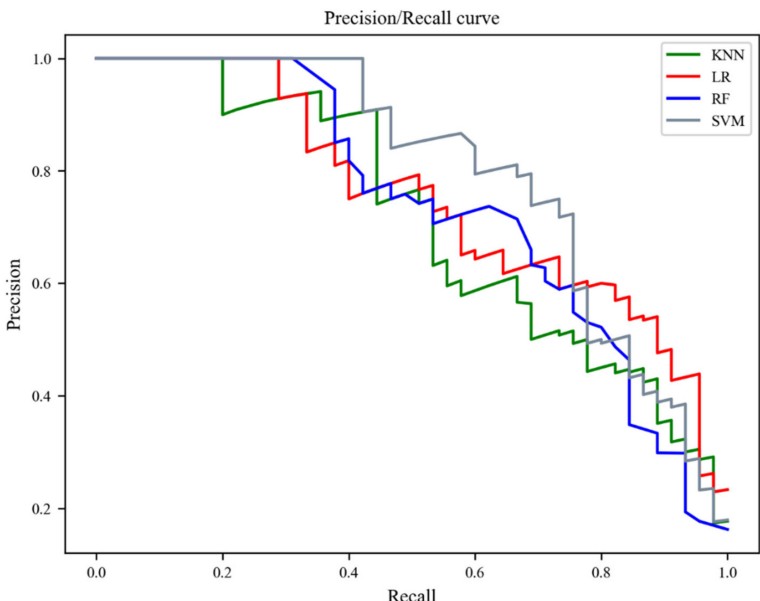

**Figure 5.** Precision–recall curves of KNN, LR, RF, and SVM on sample data after being filtered by multi-source semantic information.

**Table 3.** Precision, recall, and F1-score of KNN, LR, RF, and SVM on sample data after being filtered by multi-source semantic information.

| Classification Algorithm | Precision | Recall | F1-Score |
|:---:|:---:|:---:|:---:|
| KNN | 76.9% | 44.4% | 56.3% |
| LR | 81.0% | 37.8% | 51.5% |
| RF | 94.4% | 37.8% | 54.0% |
| SVM | 95.0% | 42.2% | 58.5% |

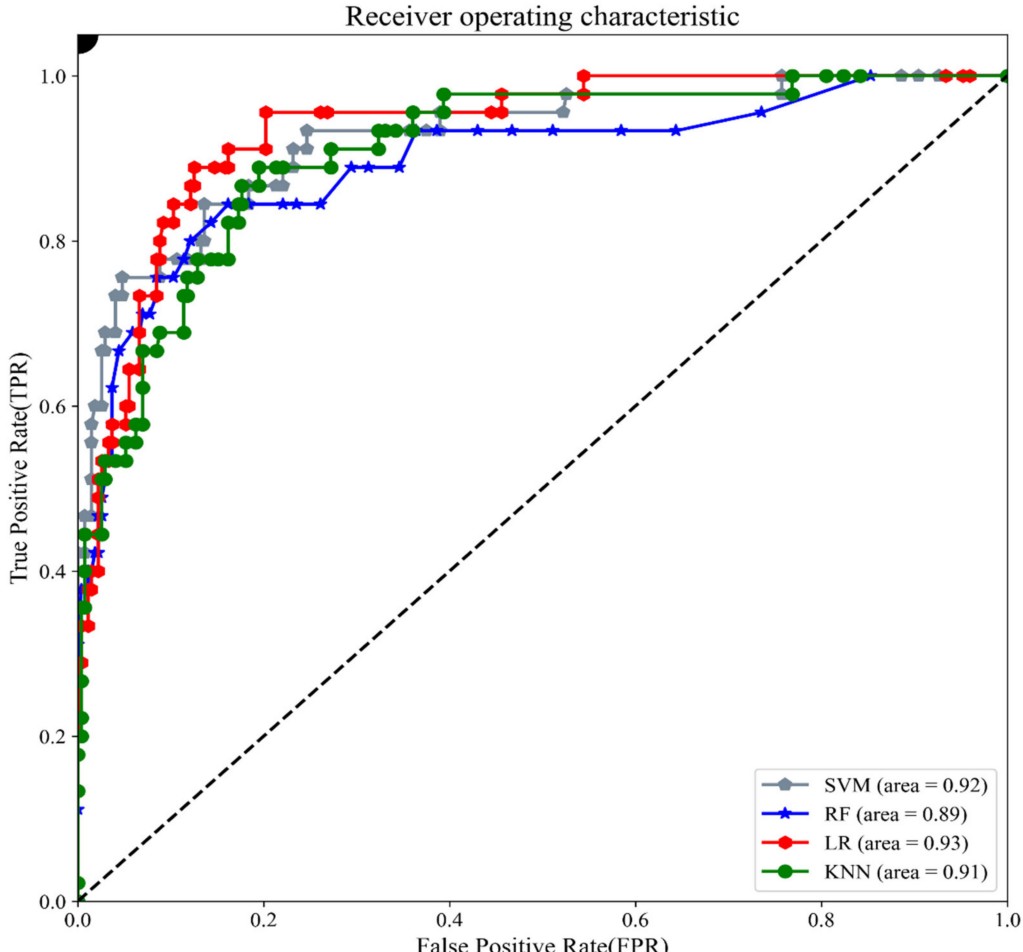

**Figure 6.** ROC curves of KNN, LR, RF, and SVM on sample data after being filtered by multi-source semantic information.

### 3.2. Robustness

To fully verify the robustness and generalizability of the proposed method, we selected keywords from the data classification system of the National Earth System Science Data Center (http://www.geodata.cn/, last accessed: 15 September 2021), which are different from those keywords selected in the literature [32] and describe the data in finer granularity. According to the method in Section 2, we obtained a total of 81,499 URLs, and obtained 1852 original sample data (called robust data) by domain name extraction and deduplication. Then, we obtained 921 sample data by filtering the 1852 initial sample data with multi-source semantic information. Finally, we applied the method in Section 3.1 to train and evaluate the robustness.

As illustrated in Table 4, when we used the data filtered with multi-source semantic information as the sample data, SVM had the highest classification precision and F1-score, which were 84.2% and 78%, respectively. Overall, the classification precision decreased compared with the value in Section 3.1, but the corresponding recall and F1-score were improved. SVM was still able to identify GDSW more precisely, which means the GDSW were identified with better robustness using the SVM method.

**Table 4.** Precision, recall, and F1-score of KNN, LR, RF, and SVM on robust data after being filtered by multi-source semantic information.

| Classification Algorithm | Precision | Recall | F1-score |
|:---:|:---:|:---:|:---:|
| KNN | 72.7% | 71.7% | 72.4% |
| LR | 53.0% | 81.8% | 64.3% |
| RF | 64.0% | 72.7% | 68.1% |
| SVM | 84.2% | 72.7% | 78.0% |

In the PR curve, as shown in Figure 7, the classification precision of SVM is higher than that of KNN, LR, and RF, while all four classification algorithms can precisely identify GDSW when recall is less than 0.6. When recall increases again, the precision of all four classification algorithms becomes a gradient downward trend. Combined with the ROC curve, as shown in Figure 8, the four classification algorithms have better classification performance on the datasets, and their AUROC values are all greater than 0.9. In other words, in this case, when the value of AUROC does not differ much, we usually select the classification algorithm with the highest classification precision as the method to identify GDSW, that is, SVM, which is also in line with the conclusion in Section 3.1.

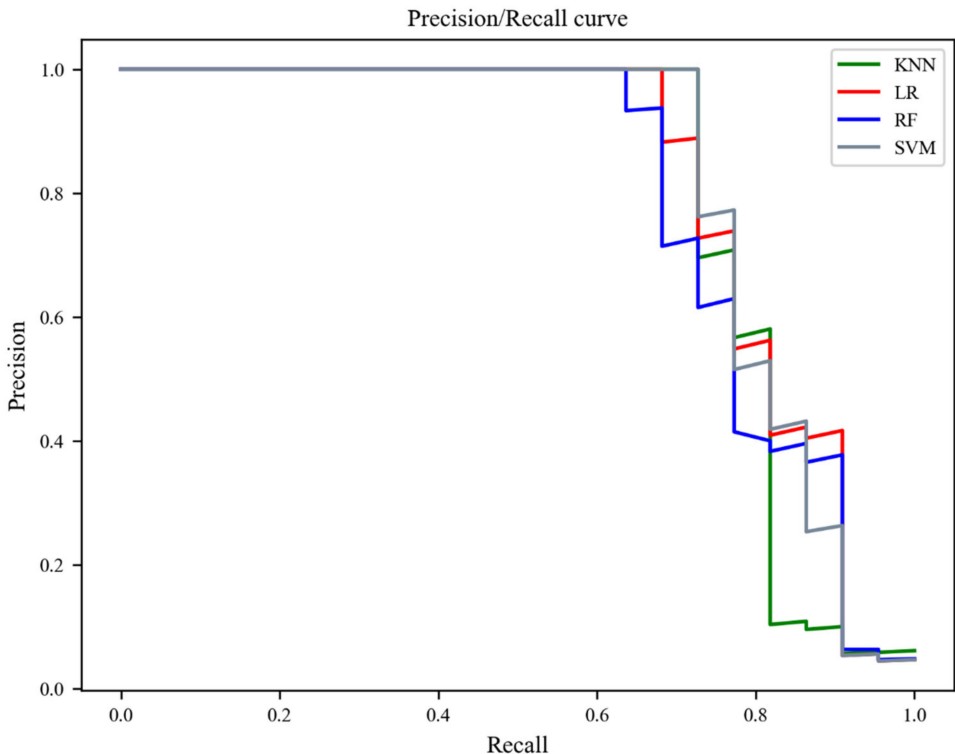

**Figure 7.** Precision–recall curves of KNN, LR, RF, and SVM on robust data after being filtered by multi-source semantic information.

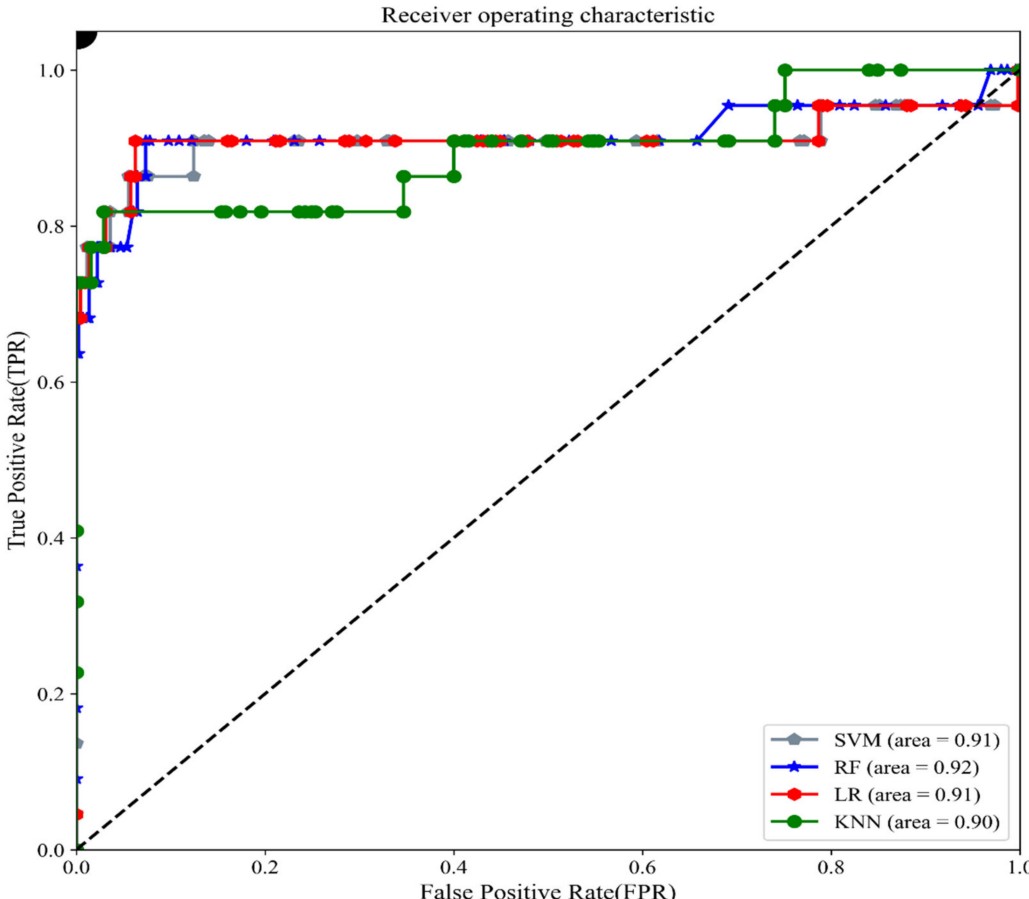

**Figure 8.** ROC curves of KNN, LR, RF, and SVM on robust data after being filtered by multi-source semantic information.

## 4. Discussion and Conclusions

This paper presents a classification method of GDSW based on machine learning. We collected a total of 79,402 websites related to geospatial data and used the proposed method to identify GDSW. Before identification, we filtered the websites by multi-source semantic information and then took the filtered websites as the sample data of machine learning. We used four classification methods to classify and predict the sample data. First, the experiment compared the effects of using a multi-source semantic similarity filter. Then, we verified the robustness of the proposed method using the same method and different data sources. The results show that the data filtered with multi-source semantic information as the sample data can identify GDSW more precisely than the pre-processed sample data only. Among the four machine learning classification methods, SVM has the highest precision in identifying GDSW. In addition, the robustness and scalability of this research are demonstrated. The results of applying the methods proposed in the paper to different sample data show that, among the four classification methods, SVM also performs best in precisely identifying GDSW among the data filtered with multi-source semantic information as the sample data for machine learning, and the classifier has higher classification performance.

There are several methodological implications in this study. First, compared with traditional methods, the identification of GDSW based on the machine learning method has significant advantages. To the best of our knowledge, there is no quick and precise method to identify GDSW at present. This method solves the problems of the existing time- and energy-consuming search based on search engine and manual assessment, and the problem that the semantic discovery is limited by existing ontology. Second, in the preparation of sample data for the precise identification of GDSW using machine learning methods, the

feasibility and usefulness of using multi-source semantic information to filter samples for improving the classification precision are demonstrated. The advantages of four machine learning algorithms for the precise identification of GDSW have been testified; specifically, SVM has the highest precision in identifying GDSW.

The methods used in this study can precisely identify GDSW, but the following issues need to be further studied in the future: (1) the influence of keywords for search. Different keywords have various search results, and the searched data are critical in GDSW mining. In the next step, we will consider more keywords as input information for search engines. (2) The selection of an ontology library. Diverse ontology libraries will lead to different comprehensive similarity values and thus different sample data. We will consider the alignment of ontology libraries in the next step to enrich the ontology libraries. (3) Due to the different forms of geospatial data publishing, the method proposed in this paper has some limitations. For example, some websites not only provide geospatial datasets but also provide a large amount of information related to institutions, which may be filtered out when filtering with multi-source semantic information. Nevertheless, some proposed methods can be used for improvement, such as website key object extraction instead of extracting website keywords based on algorithms. (4) At present, only GDSW in Chinese have been identified in the study, and GDSW provided in English and other languages will be identified in the future. The purpose is to dig out the catalog of global GDSW and comprehensively evaluate the sharing degree and quality of global geospatial data. (5) In the future, we will explore the application of existing neural network models in GDSW classification, including convolutional neural networks (CNN) [53], and vector representation models for text, including bidirectional encoder representation from transformers (Bert) [54].

**Author Contributions:** Conceptualization, Methodology, Validation, Formal Analysis, Q.C. and Y.Z.; Software, Q.C., H.Z. and L.Q.; supervision, J.S., S.W. and J.Z.; Funding Acquisition, Y.Z.; Writing—original draft preparation, Q.C.; Writing—review and editing, Q.C., Y.Z. and Y.Q. All authors have read and agreed to the published version of the manuscript.

**Funding:** This research was funded by National Natural Science Foundation of China (grant numbers: 42050101, 41771430, 41631177) and the Strategic Priority Research Program of the Chinese Academy of Sciences, (grant number: XDA23100100).

**Institutional Review Board Statement:** Not applicable.

**Informed Consent Statement:** Not applicable.

**Data Availability Statement:** The data presented in this study are available in Section 2.2.

**Conflicts of Interest:** The authors declare no conflict of interest.

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
