# Peer review of "A Method for Identifying Geospatial Data Sharing Websites by Combining Multi-Source Semantic Information and Machine Learning"

_applsci, doi:10.3390/app11188705_

Round 1

Reviewer 1 Report

See attachment

Author Response

Dear Editors and Reviewers:
    On behalf of my co-authors, we thank you very much for giving us an opportunity to revise our manuscript, we appreciate the editor and reviewers very much for their positive and constructive comments and suggestions on our manuscript. We have studied the reviewer’s comments carefully responded to them one by one, please see the attachment.
   We appreciate your comments on our manuscript. We look forward to hearing from you!

Thank you and best regards.

Yours sincerely,

Yunqiang Zhu

Corresponding author: Prof. Yunqiang Zhu

E-mail: zhuyq@lreis.ac.cn

Reviewer 2 Report

The article describes a method to identify Geospatial Data Sharing Websites (GDSW) by combining multi-source semantic information and machine learning. The methodology seems to be in order, but I wonder if this should be published in a scientific journal: it is merely a method to identify if websites contain geospatial data; it does not do anything with the scientific geospatial data itself. Analyzing only Chinese websites also makes it difficult to check for non-Chinese researchers.

Furthermore I have some minor corrections:

  • 23: website ... contains -> websites ... contain (?)
  • 220: geo-spatial -> geospatial
  • 343: "The formula is as follows:" -> where is the formula?
  • 649: Convo-lutional -> Convolutional

Author Response

(The authors gave the same response as above.)

Reviewer 3 Report

General Comments 

The work presented in this article has essentially two parts. The first part deals with building the dataset intended to be used in the second part, ie applying some ML algorithms to classify the websites (GDSW / noGDSW).

The first part needs a strong rewrite as it is quite confusing. It needs more rigor and formalization from the beginning and not just in the presentation of the FilterSim algorithm.

The 79,402 URLs that result from a search using (70 keywords) in a search engine (not specified) are from 2760 different websites (L1). Applying filtering based on "multi-source semantic information" gets 638 websites (L2)

L1 websites are manually classified into (GDSW / noGDSW) - The proportion of GDSW websites neither in L1 nor L2 nor whether these proportions are similar or different is indicated.

For these data in L1 apply 4 ML algorithms (3 runs in 50% for training and 50% for test). Results are average. They now apply with L2 data (although they evaluate with test data from the previous experience) and get slightly better results but still with F1-Score of the best algorithm with 57.1%

They apply another dataset (another search with other keywords) and surprisingly the results improve to 78% F1-Score for the best algorithm. Despite claiming that it is evidence of the proposal's robustness, this clear improvement in this new set raises more questions than it validates the proposal's robustness.

One aspect that seems very important to me is to know exactly how efficient the proposed filtering algorithm is. In other words, what is the variation in the GDSW rate with filtering and what is the final value of this rate after filtering. This information applied from various sets.

The ML part results from the basic application of the proposed techniques. I would like to see other indicators such as Kappa.

It seems to me that this work is still at a preliminary stage and needs a lot of sensitivity analysis (some aspects indicated in future work)

Few Questions / Comments 

Q: What do you mean by “.., and finally getting the key information of each website” (lines 131 - 133)?

Q: Which Search Engine? Why do you use the term fuzzy? Do you mean that it consider inexact match?

C: It would be nice to have an annex with the selected (70) keywords

C: In Figure 2 you may use keyword instead of keywords 

Q: In the crawling process you consider only the URLs embedded in the first pages of the list L.U?

Q: What do you do with the domain (URL) information 

Q: Which languages are considered in the search? Only Chinese? Also English? Other languages?

C: From 173 to 179 too much implementation details.

C: Section 2.2 is a bit confusing. Specific aspects are (unnecessarily) mixed with general background.

Q: Why naming the head content of pages as theme? 

C: There is some repetition between sections 2 and 3.1 and that doesn't always convey the same idea. Section 3.1 is confusing. 

C: The Figure 4 does not help the text and the text in the lines 267 and 273 needs a major rewrite.

Q: How do you decide the list of domains in un_u (Customized domain name libraries uncorrelated to GDSW). What is the impact of this list in the filtered Website lists? 

Calculation Method:

C: Many formatting problems, including the used width. Major review is required.

C: Notation could be improved

L - I think that instead of saying that ia Wesite listings could just “a series of URLs based on keywords crawled by search engine”

Do you really need w (Text that represents the website)? 

What do you mean by “Theme thesaurus”. It only appears in this definitions 

The step “if url in ?? :” is odd. Url is a member of the list un_u. li is a member of L_dm. 

Also it is really necessary to iterate over each url (in un_u) for each li?

Author Response

(The authors gave the same response as above.)
